# Drug-Drug Interactions Leading to Adverse Drug Reactions with Rivaroxaban: A Systematic Review of the Literature and Analysis of VigiBase

**DOI:** 10.3390/jpm11040250

**Published:** 2021-03-30

**Authors:** Silvia Fernandez, Camille Lenoir, Caroline Flora Samer, Victoria Rollason

**Affiliations:** 1Department of Anesthesiology, Pharmacology, Intensive Care and Emergency Medicine, Division of Clinical Pharmacology and Toxicology, Geneva University Hospitals, 1205 Geneva, Switzerland; silviapfm@gmail.com (S.F.); caroline.samer@hcuge.ch (C.F.S.); 2Institute of Pharmaceutical Sciences of Western Switzerland (ISPSO), University of Geneva, 1206 Geneva, Switzerland

**Keywords:** rivaroxaban, drug-drug interactions, pharmacokinetic, adverse drug reaction, spontaneous reports

## Abstract

Rivaroxaban has become an alternative to vitamin K antagonists, which are considered to be at higher risk of drug-drug interactions (DDI) and more difficult to use. However, DDI do occur. We systematically reviewed studies that evaluated them and analysed DDI and subsequent adverse drug reactions (ADR) reported in spontaneous reports and VigiBase. We systematically searched articles that explored DDI with rivaroxaban up to 20 August 2018 via Medline, Embase and Google Scholar. Data from VigiBase came from spontaneous reports recovered up to 2 January 2018, where Omega was used to detect signals and identify potential interactions in terms of triplets with two drugs and one ADR. We identified 31 studies and 28 case reports. Studies showed significant variation in the pharmacokinetic for rivaroxaban, and an increased risk of haemorrhage or thromboembolic events due to DDI was highlighted in case reports. From VigiBase, a total of 21,261 triplets were analysed and the most reported was rivaroxaban–aspirin–gastrointestinal haemorrhage. In VigiBase, only 34.8% of the DDI reported were described or understood, and most were pharmacodynamic DDI. These data suggest that rivaroxaban should be considered to have significant potential for DDI, especially with CYP3A/P-gp modulators or with drugs that impair haemostasis.

## 1. Introduction

Unlike heparin or vitamin K antagonists (VKAs), direct oral anticoagulants (DOACs) act by direct inhibition of coagulation factor Xa or factor II (thrombin). Their pharmacological profile is deemed predictable, safe and suitable for long-term use [1]. While VKAs were the only available oral anticoagulants for more than 50 years, clinical requirements for a variety of indications in adults and the willingness to have new antithrombotic drugs on hand with an optimal balance between efficacy and risk of bleeding led to the emergence of DOACs [2]. Indeed, DOACs are considered easier to use because they have a wide therapeutic window, less interindividual variability, and higher oral bioavailability that is less impacted by food intake or bodyweight than warfarin, the reference treatment [3]. Therefore, they no longer needed to be individualised on a daily basis like VKAs, which require monitoring of the international normalised ratio (INR) [3]. However, although DOACs are less influenced by food or bodyweight, small dose adjustments are necessary for a high-dose of rivaroxaban and for low-weight patients < 60 kg taking apixaban respectively [3].

There are currently five DOACs approved for use worldwide: dabigatran, an oral direct thrombin (factor II) inhibitor [4]; rivaroxaban; apixaban; edoxaban, and; betrixaban [5], which are oral direct factor Xa inhibitors [5].

Rivaroxaban was the first oral direct factor Xa inhibitor approved, and it is used to prevent and/or treat venous thromboembolism (VTE) and prevent the occurrence of ischaemic stroke and embolism in individuals with nonvalvular atrial fibrillation (NVAF) [2]. In patients with NVAF and acute symptomatic VTE, studies have demonstrated that rivaroxaban is as effective as the standard therapy [6,7,8]. Moreover, rivaroxaban was superior to enoxaparin for the prevention of VTE in patients undergoing major orthopaedic surgery [9,10,11,12]. There is, therefore, no additional need for a priori monitoring of specific laboratory parameters, but anti-Xa factor could be used in specific cases where measurement is needed, for example, to confirm an overdose [13].

In addition to its ease of use and efficacy, rivaroxaban is considered to have a low risk of drug-drug interactions (DDIs), although two-thirds of rivaroxaban elimination takes place via conversion to inactive metabolites in the liver by CYP3A [3]. Rivaroxaban also carries an inherent risk of bleeding, and its coadministration with other drugs affecting haemostasis can lead to an increased risk of haemorrhage [14].

Like all DOACs, rivaroxaban has certain limitations in its use [15]. Rivaroxaban is contraindicated in women during pregnancy and lactation and in children because no data are available for these populations [16]. In addition, rivaroxaban should not be prescribed in patients with severe hepatic (Child Pugh C), renal impairment (creatinine clearance < 15 mL/min), antiphospholipid syndrome or mechanical heart valves [16]. No dose adjustments are recommended for rivaroxaban based on sex, age or bodyweight [17].

Regarding the safety profile, patients receiving rivaroxaban for any therapeutic indication have a lower risk of intracranial bleeding compared to patients receiving VKAs alone or in sequential treatment with low-molecular-weight heparins [18]. However, gastrointestinal bleeding seems to be more frequent [19,20]. Bleeding is not the only safety concern with rivaroxaban, as it has been associated with a risk of hepatotoxicity in a review that analysed data from case reports, case series and spontaneous reports [20,21,22]. However, in more than one-third of the drug-induced liver injuries (DILIs) observed, concomitant use of possible hepatotoxic and/or interacting drugs was also reported [21,22]. Based on these results, the authors suggested that there is a need to re-evaluate the risk of DILI associated with rivaroxaban and the importance of post-marketing pharmacovigilance to detect these potential adverse drug reactions (ADRs) [21,22].

The global aim of this study was to evaluate DDIs causing ADRs with rivaroxaban through a review of currently published data in the literature and a real-world evaluation of rivaroxaban’s interaction data from VigiBase, the WHO (World Health Organization) global database of individual case safety reports (https://www.who-umc.org) [23].

## 2. Materials and Methods

### 2.1. Literature Search in Biomedical Databases

As suggested by the Preferred Reporting Items for Systematic Review and Meta-Analyses (PRISMA) Statement, the eligibility criteria were divided into two key categories [24]. The eligibility criteria were applied to select relevant publications and are described in Table 1 [25]. The literature search was done for articles published up to 20 August 2018 in Google Scholar and in two databases, specifically, Embase and PubMed via MEDLINE. The literature search was achieved for four DOACs (rivaroxaban, apixaban [25], edoxaban and dabigatran), and the search approach was developed independently for Google Scholar, Embase and PubMed, as previously described [25]. For Google Scholar, the keywords/strings were rivaroxaban OR apixaban OR dabigatran OR edoxaban AND interaction OR interactions AND “case report”. For Embase, the keywords/strings used were (rivaroxaban OR apixaban OR dabigatran OR edoxaban) OR (DOACs OR NOAC OR “direct oral anticoagulants” OR “new oral anticoagulants” OR “direct thrombin inhibitor” OR “direct factor Xa inhibitor”) AND drug interaction. Finally, for PubMed, the keywords/strings used were (rivaroxaban OR apixaban OR dabigatran OR edoxaban) OR (DOACs OR NOAC OR “direct oral anticoagulants” OR “new oral anticoagulants” OR “direct thrombin inhibitor” OR “direct factor Xa inhibitor”) AND (drug interaction OR interaction).

The reference managing software Zotero^®^ (version 5.0.47) was used to remove duplicates, and the potential relevance of the remaining records was assessed by two reviewers who screened the title and abstract. If a single study was described in more than one article and each presented the same data, the most recent study was integrated. The included articles were divided into two groups, namely, interaction studies and case reports.

The mechanisms of interactions for interaction studies and case reports were checked by reviewing the Summary of Product Characteristics (SmPC) [14], UpToDate-Lexicomp [26] and the CYP450 substrates, inhibitor and inducers table (https://www.hug-ge.ch/sites/interhug/files/structures/pharmacologie_et_toxicologie_cliniques/a5_cytochromes_6_2.pdf (accessed on 20 August 2018)) [27]. Case reports that described DDIs that were not previously described or not understood from a pharmacological point of view were excluded.

As already done with apixaban in a previous article, the types of interactions assessed were PK interactions mediated by CYP3A, P-gp modulators and/or by gastric pH modifiers and PD interactions mediated by other antithrombotic agents and nonsteroidal anti-inflammatory drugs (NSAIDs) for interaction studies [25]. An additional category entitled “other drugs” pooled interactions not matching any of the previous categories. Each study was reviewed and described individually and categorised into in vitro/animal studies or phase I to phase IV human studies. Furthermore, a post hoc analysis was performed to allow us to assess if some DDIs were missing and if the SmPC included all the DDIs described in the literature.

Concerning case reports, the required information was patient characteristics, information on rivaroxaban (dosage, start and end of treatment, duration of treatment) and potential interacting drugs, adverse event descriptions and a list of additional medication when available.

### 2.2. Analysis of Data from Spontaneous Reports in VigiBase

Spontaneous reports from VigiBase were used to investigate DDIs between rivaroxaban and other drugs. The Uppsala Monitoring Centre (UMC) is the WHO Collaborating Centre for International Drug Monitoring and is responsible to maintain VigiBase. UMC receives reports of suspected ADRs from national centres in countries participating in the WHO Program for International Drug Monitoring (https://www.who-umc.org/vigibase/vigibase/ (accessed on 2 January 2018)). At the date of retrieval (accessed on 2 January 2018), there were a total of 16,329,758 individual case safety reports (ICSRs) in VigiBase that came from 131 countries. Drugs are coded according to WHODrug and ADRs are coded according to MedDRA (version 20.1) [28]. The information in VigiBase comes from multiple sources, and the probability that the suspected adverse effect is drug-related is not the same in all cases [29].

Each ICSR retrieved contained drug-drug-ADR (DDA) triplets that allowed the identification of potential DDIs. The analyses to detect potential signals of DDIs were performed using Omega (Ω), an observed-to expected three-way measure of disproportionate reporting developed by the UMC [30]. When Ω is positive and two drugs are used together, an increased risk of a specific ADR occurrence is emphasised over the sum of the individual risks when these same drugs are used separately. Thus, it is an indicator of the frequency of reporting of certain DDA triplets in the dataset compared to what is expected based on the relative reporting in the dataset. The Ω value is thereby dynamic because it can change as new reports are entered in VigiBase. Ω_0.25_ is used as a threshold in the screening of potential DDIs in data from ICSRs because it is the lower limit of a 95% credibility interval for Ω. Prior to analysis, the data set was thus cleaned by removing all DDAs with Ω_0.25_ less than or equal to 0. The next step to clean the data set was to exclude some non-relevant MedDRA preferred terms, such as “stent placement”, “vascular stent insertion” and “Dieulafoy’s vascular malformation”. Some non-relevant drug names were also excluded. Finally, all rows with drugs reported as “concomitant” were removed from the file; therefore, only drugs reported as “interacting” or “suspected” were kept. This cleaning procedure was the same as that already described in a previous publication [25].

The search and extraction of ICSRs related to rivaroxaban and DDIs from VigiBase were performed by the UMC on 24 April 2018 from a database freeze conducted on 2 January 2018 [25]. The number of DDA triplets for rivaroxaban related to each MedDRA system organ class (SOC), the number of DDA triplets for rivaroxaban and one specific ADR and the number of combinations for rivaroxaban and one specific suspected/interacting drug in the DDA triplet were studied.

According to SmPC, UpToDate and PubMed, DDIs were classified in PK and/or PD DDIs and in unknown DDIs. PK and PD DDIs were further classified into sub-classifications that included absorption (PKA), distribution (PKD), metabolism (PKM) or excretion (PKE) for PK mechanisms and direct effects on receptor function (PD1), interference with a biological physiological control process (PD2) or additive/opposed pharmacological effects (PD3) for PD mechanisms. DDIs were counted in when they were verified for the two mechanisms. All mechanisms were listed when more than one was found. This article focuses on rivaroxaban only, due to the large quantity of data extracted with the VigiBase analysis. As already mentioned, similar work was done for apixaban only [25].

## 3. Results

### 3.1. Literature

The literature search retrieved 31 interaction studies, some investigating several drugs, and 28 case reports. The selection process is illustrated in the following PRISMA flowchart (Figure 1).

#### 3.1.1. CYP3A and P-gp Inhibitors

##### In Vitro Studies

Rivaroxaban did not show any interaction with tacrolimus when both drugs were supplemented into citrated plasma in an in vitro study [31]. In vitro, type 5 phosphodiesterase inhibitors (PDE5is), such as sildenafil, tadalafil and vardenafil, inhibited the P-gp-mediated efflux of rivaroxaban [32]. According to the authors, this could have consequences on rivaroxaban’s safety, particularly in terms of bleeding risk [32].

##### Phase I Studies

In healthy volunteers, coadministration of ketoconazole increased the rivaroxaban AUC by 158% and the Cmax by 72% [33]. Similarly, ritonavir significantly increased the rivaroxaban AUC and Cmax by 153% and 55%, respectively [33]. Coadministration of clarithromycin, erythromycin and fluconazole with rivaroxaban significantly increased its AUC by 54%, 34% and 42%, respectively, but these moderate effects were not considered to be clinically relevant [33]. All of these coadministered drugs are CYP3A/P-gp inhibitors.

Another phase I study found a high impact of clarithromycin on rivaroxaban’s PK with an AUC increase of 94% and a Cmax increase of 92%, independent of the ABCB1 genotype [34]. The effect of erythromycin on rivaroxaban exposure was also assessed in another study but this time in subjects with normal and impaired renal function. In subjects with normal renal function, coadministration with erythromycin produced an increase in the rivaroxaban AUC and Cmax of 39% and 40%, respectively [35]. However, in subjects with mild renal impairment, the increase in the rivaroxaban AUC and Cmax when given erythromycin was 54% and 26%, respectively. Moderate renal impairment combined with erythromycin coadministration increased rivaroxaban’s AUC and Cmax by 71% and 21%, respectively [35]. A study conducted in healthy volunteers demonstrated that verapamil coadministration increased the AUC of rivaroxaban. Volunteers were separated into a normal renal function group and a mild renal impairment group. The increase in the AUC was of the same extent in both groups (ratio of geometric means: 1.39 vs. 1.43, respectively) [36].

##### Phase II Studies

Limited data from a small phase II study in liver transplant patients (*n* = 9) showed an increase in rivaroxaban plasma levels in the presence of cyclosporin A (*n* = 5) [37]. The rivaroxaban plasma levels were within therapeutic ranges in patients treated with tacrolimus instead of cyclosporin A [37]. In a study that compared patients taking rivaroxaban (controls) and patients taking rivaroxaban and diltiazem, there was no significant difference in the incidence of major and/or clinically relevant non-major bleeding events in either group [38]. The authors suggest that although diltiazem may increase rivaroxaban exposure because of its moderate inhibition of CYP3A/P-gp, there was no evidence of an increased risk of bleeding outcomes in patients taking both drugs [38].

##### Phase III Studies

A study that used data from the ROCKET study assessed the risk of coadministration of non-dihydropyridine calcium channel blockers (non-DHP CCBs) with rivaroxaban or warfarin. This coadministration was not associated with a significant increase in the risk of stroke or non-central nervous system systemic embolism (*p* = 0.11) or major or non-major clinically relevant bleeding (*p* = 0.087) [39]. However, major bleeding or intracranial haemorrhage occurred more frequently in the non-DHP CCB user group (*p* = 0.0091 and *p* = 0.001, respectively) [39]. Cardiovascular death, all-cause death, myocardial infarction and all-cause hospitalisation were not significantly different between the two groups [39]. Comparison between rivaroxaban and warfarin users showed no significant difference in safety outcomes such as major or non-major clinically relevant bleeding (*p* = 0.14) in non-DHP CCB users [39].

##### Phase IV Studies

A retrospective study concluded that coadministration of amiodarone and rivaroxaban is linked to an increased risk of bleeding [40]. The study compared the number of bleeding events in patients being treated simultaneously with both drugs with patients taking rivaroxaban only [40]. Another retrospective study assessed the bleeding risk of rivaroxaban and other DOACs when it was coadministered with verapamil, diltiazem, amiodarone, dronedarone, azoles (fluconazole, ketoconazole, itraconazole, voriconazole and posaconazole), cyclosporine, erythromycin or clarithromycin [41]. The combination of rivaroxaban with amiodarone and fluconazole was associated with a significantly increased risk of major bleeding [41]. In contrast, the coadministration of rivaroxaban and erythromycin or clarithromycin decreased the risk of major bleeding but it was not statistically significant [41]. Coadministration of rivaroxaban with cyclosporin, verapamil, diltiazem, ketoconazole and itraconazole, voriconazole, posaconazole and dronedarone did not significantly change the incidence rate of major bleeding [41]. Results for erythromycin, clarithromycin, cyclosporin, verapamil, ketoconazole and dronedarone are not in line with previously cited studies. This could be because this study has strong limitations of a retrospective design and of an analysis based on the Health Insurance database system, and thus, has a lack of detailed clinical information such as liver and kidney function [41]. Moreover, this study included an Asian population that has been recognised to have a different bleeding risk and anticoagulant therapy than the Western population [41]. Finally, rivaroxaban dosage and concomitant treatment were not considered in the model [41].

##### In Silico Studies

A study combined data from in vitro inhibition assays and static modelling to predict in vivo DDIs between rivaroxaban and amiodarone and dronedarone, two CYP3A/P-gp inhibitors. Thus, the study predicted an increased rivaroxaban exposure of 37% and 31%, respectively [42]. In addition, a nine percent increase in rivaroxaban exposure due to inhibition of P-gp-mediated efflux by either of the two inhibitors was estimated [42]. In a study that developed a physiologically based pharmacokinetic (PBPK) model, rivaroxaban exposure increased when DDIs with CYP3A/P-gp inhibitors (ketoconazole, ritonavir, clarithromycin) coexisted with mild or moderate hepatic dysfunction compared to hepatic dysfunction alone [43]. The simulation results revealed a synergistic effect of the addition of DDI and hepatic dysfunction, which was greatest when hepatic dysfunction was severe [43]. Another PBPK study showed that coadministration of verapamil and rivaroxaban increased rivaroxaban AUC by 1.48-fold and that coadministration of verapamil and renal insufficiency produced a synergistic increase in systemic exposure to rivaroxaban [44]. The authors suggested that subjects with mild to severe renal dysfunction who are taking verapamil should receive a reduced dose of rivaroxaban to minimise synergistic drug-drug disease interactions and prevent further bleeding risks [44]. In another PBPK model, systemic exposure to 20 mg of rivaroxaban once daily for 20 days increased when coadministered with a loading dose of amiodarone 200 mg three times a day during the last fifteen days in healthy subjects [45]. Simulations also indicated a significant 1.36-fold mean AUC increase [45]. Moreover, renal insufficiency had a synergistic effect, as the simulated mean AUC-fold change was 1.86- in patients with mild renal impairment and 1.61 in patients with moderate renal impairment where the rivaroxaban dosage was reduced to 15 mg [45].

#### 3.1.2. CYP3A and P-gp Inducers

##### Phase IV Study

Coadministration of rivaroxaban with rifampicin and phenytoin was assessed and surprisingly showed an increased risk of major bleeding [41]. However, this effect was not statistically significant for rifampicin [41]. Phenytoin, as a CYP inducer, is expected to decrease rivaroxaban AUC and, therefore, the bleeding risk. The results of this phase IV study should be treated with caution due to the limitations mentioned above [41].

#### 3.1.3. CYP3A and P-gp Substrates

##### Phase I Studies

No clinically relevant PK or PD interactions between rivaroxaban and the CYP3A substrate midazolam, the P-gp substrate digoxin and the CYP3A/P-gp substrate atorvastatin were observed in healthy volunteers [33,46].

##### Phase IV Study

The bleeding risk with rivaroxaban was assessed when coadministered with atorvastatin and digoxin and a significantly decreased risk of major bleeding was observed, while the effect of digoxin was not statistically significant [41]. In the phase I study, atorvastatin had no effect on rivaroxaban PK and this discrepancy in results can also be attributed to the limitations of the phase IV study [41].

#### 3.1.4. Other Antithrombotic Agents and NSAIDs

##### In Vitro and Animal Studies

The combination of rivaroxaban with acetylsalicylic acid (ASA) and/or ticagrelor in vitro using human platelet-rich plasma and coadministration of low-dose rivaroxaban with ASA and clopidogrel in rat models of arterial thrombosis suggested that the combination of rivaroxaban with single or dual antiplatelet agents led to a synergistic increase in their antithrombotic activity compared with anticoagulant or antiplatelet therapy alone [47]. Furthermore, the authors considered that since the low dose of rivaroxaban tested was equivalent to the trough plasma concentration after a rivaroxaban 2.5 mg twice daily dose in humans, their results can be deemed of clinical relevance [47].

#### Phase I Studies

No clinically relevant PK interactions were observed between rivaroxaban and enoxaparin [48] or warfarin in phase I studies [49,50]. However, some rivaroxaban PD parameters were affected, and the anti-factor Xa activity of rivaroxaban increased by 50% when coadministered with enoxaparin [48]. Regarding warfarin, an additive effect on the prolongation of the PT/INR was observed during the initial transitioning period from warfarin to rivaroxaban, although pre-treatment with warfarin did not affect rivaroxaban anti-factor Xa activity [49]. Similar results arose during the co-treatment period when switching from rivaroxaban to warfarin (higher PT and greater than additive INR values than those measured when either drug was administered alone) [50]. The combination of rivaroxaban and the commonly used NSAID naproxen significantly increased the bleeding time compared with rivaroxaban alone. On the other hand, rivaroxaban exposure was only slightly affected by coadministration of both drugs (10% increase in the rivaroxaban AUC and Cmax). The authors of the study concluded that there was no clinically relevant interaction between rivaroxaban and naproxen [51]. Moreover, the same finding was found for rivaroxaban and ASA. Rivaroxaban’s bleeding time was prolonged when both drugs were coadministered as compared to rivaroxaban alone, while its PK characteristics/properties remained unchanged. Thus, the authors considered that the rivaroxaban-ASA interaction was not clinically relevant [52]. Coadministration of rivaroxaban and clopidogrel led to an additive effect on the bleeding time that was doubled when compared with the effect produced with clopidogrel alone, without affecting any other PK or PD parameters of rivaroxaban [53].

##### Phase II Studies

In acute coronary syndrome (ACS) patients, rivaroxaban increased the risk of bleeding events in a dose-dependent manner in both groups of patients (aspirin or aspirin and thienopyridine) compared to placebo [54]. Moreover, the absolute rate of clinically significant bleeding was higher in the group receiving dual antiplatelet therapy than in the group receiving ASA alone in addition to rivaroxaban [54]. In a study that compared the use of a low dose of rivaroxaban (2.5 mg twice daily) concomitant with either clopidogrel or ticagrelor to dual antiplatelet therapy (aspirin and either clopidogrel or ticagrelor) in patients who underwent percutaneous coronary intervention, there were no significant differences in the bleeding incidence [55]. However, in a post hoc analysis, the use of ticagrelor was associated with a significant increase in the bleeding rate (*p* = 0.0006) compared to clopidogrel [55]. As pointed out by the authors, a higher bleeding rate was found in regions where there was greater use of ticagrelor but was not associated with the randomised treatment assignment (rivaroxaban vs. aspirin) [55].

##### Phase III Studies

In a sub-analysis of pooled data from the RECORD programme, coadministration of NSAIDs (relative rate ratio = 1.22, CI95% = 0.99–1.50), platelet aggregation inhibitors (PAIs) and ASA (relative ratio = 1.32, CI95% = 0.85–2.05) together with rivaroxaban increased the risk of bleeding in hip or knee replacement surgery patients, although the effect was not considered significant [56]. However, the small proportion of patients using concomitant PAIs and ASA may not have been high enough to conclude on the risk of bleeding, which could explain the difference in results with other studies [56]. Regarding the increased risk of bleeding with concomitant use of NSAIDs, it was at the limit of statistical significance [56]. In the ROCKET-AF trial, more than one-third of patients were on ASA at baseline, and the concomitant use of rivaroxaban and ASA was associated with higher rates of all-cause death [57]. It is worth mentioning that the increase in all-cause death in the presence of aspirin was more pronounced for warfarin than for rivaroxaban, enhancing the difference between the two drugs regarding outcome [57].

##### Phase IV Studies

In a sub-analysis of the XAMOS study, coadministration of PAIs (including ASA) increased the incidence of symptomatic thromboembolic and bleeding events in patients taking rivaroxaban and in those who followed standard thrombophylaxis for VTE prophylaxis after major orthopaedic surgery [58]. However, this finding was largely attributable to a higher incidence of symptomatic arterial thromboembolic events [58]. This could be explained by the fact that PAIs users were older and had more comorbidities affecting cardiovascular risk [58]. Additionally, concomitant use of NSAIDs was also associated with an increased risk of bleeding, while it had no influence on the rate of symptomatic thromboembolic events [58].

#### 3.1.5. Gastric pH Modifiers

##### Phase I Studies

Ranitidine, a H_2_ antagonist, has no significant impact on the PK/PD of rivaroxaban [59]. Similarly, the proton pump inhibitor (PPI) omeprazole showed no clinically relevant PK or PD interactions with rivaroxaban [60].

#### 3.1.6. Other Drugs

##### In Vitro Studies

Irinotecan is metabolised by esterases to its active metabolite SN-38 (7-ethyl-10-hydroxycamptothecin), which is later detoxified via glucuronidation to form SN-38G. In human liver microsomes, rivaroxaban displayed dose-dependent inhibition of SN-38 glucuronidation, which may increase SN-38 toxicity [61]. These findings suggest a potential interaction between rivaroxaban and irinotecan [61]. The combination of rivaroxaban with drugs such as alendronate sodium, chondroitin sulfate, hydrocodone-acetaminophen, clonazepam, penicillin, tramadol and tranexamic acid did not exhibit any interactions [31].

Results are summarised in Table 2.

### 3.2. Case Series or Reports

Twenty-eight case reports were found in the literature. Eleven cases were female, with an overall age range of 29–90 years. Among them, four patients died. The rivaroxaban indication was mainly AF (*n* = 16) but also VTE prevention after orthopaedic surgery (*n* = 7), recurrent VTE prevention (*n* = 2), VTE treatment (*n* = 1), transient ischaemic attack (*n* = 1) and unknown (*n* = 1). Renal impairment (*n* = 7) was the most relevant pathophysiological factor contributing to the development of ADRs. Concerning the mechanism of interaction, PK DDIs were involved in seventeen cases [63,64,65,66,67,68,69,70,71,72,73,74,75,76,77,78,79], PD DDIs in eight cases [80,81,82,83,84,85,86,87] and PK/PD DDIs in three cases [88,89,90]. Bleeding (*n* = 18) and thromboembolic events (*n* = 7) were the two main ADRs described in these case reports. In two other cases, the coagulation parameters were abnormal, and the anti-Factor Xa peak remained under the reference value, but this had no consequences [78,89]. In one case, rivaroxaban induced hepatic encephalopathy that led to death [90]. In the cases describing thromboembolic events or lack of efficacy measured with laboratory tests (coagulation parameters or anti-Factor Xa), the involved comedications were CYP3A and/or P-gp inducers, namely, rifampicin [68,73], nevirapine [72] and antiepileptic drugs, such as carbamazepine [64,66,77], oxcarbamazepine [65] or phenytoin [78]. PK DDIs with CYP3A and/or P-gp inhibitors led to bleeding events in all cases. The PD DDIs involved coadministration of alirocumab [80] and antiplatelet aggregation drugs such as clopidogrel [80,86] or aspirin [87], warfarin [81,85], NSAIDs [83,84] and cocaine [82].

### 3.3. VigiBase

A total of 21,261 DDAs with positive Ω_0.25_ values were extracted from VigiBase for the DDA combination of rivaroxaban with any suspected/interacting drug and any ADR. Those DDAs came from 18,928 ICSRs reported to VigiBase up to the database freeze in January 2018. After cleaning the datasets, 21,109 DDAs (corresponding to 862 unique DDA combinations of rivaroxaban with one specific suspected/interacting drug and one defined ADR, each observed in a certain number of ICSRs). In the dataset, the most represented MedDRA SOCs were GI disorders (*n* = 12,307, 58.3%), renal and urinary disorders (*n* = 1994, 9.4%) and vascular disorders (*n* = 1533, 7.3%). For the ADRs, the three most reported in combination with rivaroxaban and any other suspected/interacting drug were GI haemorrhage (*n* = 7182, 34.0%), upper GI haemorrhage (*n* = 1619, 7.7%) and rectal haemorrhage (*n* = 1355, 6.4%). Regardless of the ADR, acetylsalicylic acid (ASA) (*n* = 12,725, 60.3%), clopidogrel (*n* = 2464, 11.7%) and warfarin (*n* = 1110, 5.3%) were the three suspected/interacting drugs most co-reported with rivaroxaban. If the ADRs reported for each of those drug pairs were also considered, the most reported ADR was GI haemorrhage, with incidence rates of 38.0% (*n* = 4838), 40.9% (*n* = 1009) and 36.6% (*n* = 406), respectively.

The three most reported DDAs in the whole dataset were:rivaroxaban–ASA–GI haemorrhage (*n* = 4838, 22.9%)rivaroxaban–ASA–Upper GI haemorrhage (*n* = 1040, 4.9%)rivaroxaban–clopidogrel–GI haemorrhage (*n* = 1009, 4.8%)

Of the 862 DDAs reviewed, 559 DDIs were not verified in the literature. A total of 41 PK DDIs and 265 PD DDIs were verified in the literature. The most common PK DDI was inhibition of drug metabolism, and the most common PD DDI was additive pharmacological effects.

Concerning verified PK DDIs, inhibitors of CYP3A and P-gp were the most reported drugs, and bleeding was the most reported ADR (Table 3). Regarding verified PD DDIs, antithrombotic agents and NSAIDs were the most reported drugs, and bleeding was also the most reported ADR. Regarding bleeding, the most reported site was the gastrointestinal tract (Table 3). Table 3 shows the number of occurrences that represent the number of different ADRs that occurred after the interaction between rivaroxaban and drug B, and the number in parentheses is the number of the most frequently reported ADR.

## 4. Discussion

Due to their ease of use and alleged favorable safety and efficacy profile, anticoagulation drug management experienced a major turning point with the arrival of DOACs, especially rivaroxaban, which was the first to be marketed in 2009 for cardiovascular indications [14,91]. As rivaroxaban has been on the market for several years, it has been increasingly possible to highlight DDIs in real-world situations. In line with this, we performed a systematic review of published studies and case reports, together with an analysis of data reported to VigiBase, as already done with apixaban in a previous article [25]. We showed that rivaroxaban is subject to a significant number of DDIs that need to be considered by clinicians and patients, especially DDIs with CYP3A/P-gp inhibitors and other antithrombotic agents/NSAIDs. The impact of inducers of CYP3A/P-gp on rivaroxaban is sparsely available in the literature. A post hoc comparison between collected interactions in the literature and interactions contained in rivaroxaban’s SmPC was performed to verify the accuracy of our review [14]. First, the DDI between rivaroxaban and rifampicin reported in the rivaroxaban SmPC was not detected by our literature search and not registered in clinicaltrials.gov, which means that this study is not publicly available in any form and seems not to have even been registered in any national or international registry so far, even though registries of clinical trials are an important data source in clinical research. Conversely, some interactions that were identified by our search are not included in the SmPC. This can be explained by the fact that not all information has to be disclosed in the SmPC. Concerning in vitro interaction studies, data are only integrated into the SmPC if they impact the use of the medicinal product [61,92,93]. A lack of interaction should only be mentioned in the SmPC if it is of major significance to the prescriber for data from in vivo studies. Moreover, phase I studies in healthy volunteers publication depends on the transparency policies of drug manufacturers because they are not subject to required data disclosure [94,95]. Compared to studies performed in patients, a recent study showed that phase I (conducted in healthy volunteers) studies had a significantly lower level of transparency [95]. Finally, data from post-marketing studies are only included if they result in a variation of the drug’s marketing authorisation [93,96].

Venous thromboembolism was identified in the case reports included in our literature search as one of the most frequently reported ADR of rivaroxaban, and it was not mentioned, per se, in rivaroxaban’s SmPC. This is likely due to the fact that interactions leading to this ADR are not recognised and are instead classified as treatment inefficacy [20]. Therefore, this is not a lack of coverage in our literature search.

Regarding data from VigiBase, the most co-reported suspected/interacting drug was ASA, the most co-reported ADR was GI haemorrhage, and consequently, rivaroxaban–ASA–GI haemorrhage was the most reported DDA triplet. These results are not surprising, as multiple studies have highlighted the increased risk of GI haemorrhage when DOACs were administered, including a thorough evaluation of their safety profile based on data from the same source, VigiBase [19,20]. More precisely, rivaroxaban showed a positive odds ratio of 1.38 (1.24–1.55) for GI haemorrhage compared to warfarin [20]. Several suspected/interacting drugs were not documented or understood from a pharmacological point of view to be associated with a DDI with rivaroxaban, so they were excluded from our analysis of the ICSRs. Moreover, with the dataset available, it was not possible to find a plausible explanation for some of the DDIs, and many DDA triplets did not seem to correlate, such as rivaroxaban with mesalazine and poor-quality sleep. The data stored in VigiBase come from regulatory and voluntary sources and may lack a proper causality assessment in some cases, since not all national pharmacovigilance centres contributing to VigiBase perform causality assessments of their ICSRs [97]. Additionally, some cases may lack completeness, and the data stored are heterogeneous. Rivaroxaban might be at higher risk of interacting with drugs with the same pharmacological profile because the proportion of DDIs involving the PD mechanism was higher than the proportion of DDIs involving the PK mechanism. This finding erroneously suggests that rivaroxaban might not interact with CYP3A/P-gp inhibitors or inducers. Indeed, this emphasises a bias in the data included in VigiBase, which depends on spontaneous reporting. As healthcare professionals and/or patients are the source of these spontaneous reports and as they are often less familiar with PK DDIs, these are underreported because they go undetected. These results are consistent with those of a study that used the same database, where PD and PK DDIs accounted for 41% and 25% of DDIs, respectively [98].

VigiBase has inherent limitations, as all ADR reporting databases [99]. Underreporting and selective reporting are the two first limitations worth mentioning. Another limitation of these databases is the unfeasibility of estimating risk, due to the absence of a denominator. Using certain reporting patterns as indicators of DDIs in addition to a positive Ω_0.25_ is one of the ideas that have been put forward for improving the database [100]. The existence of a plausible time course, a positive dechallenge and alternate causes of the reaction could help identify suspected adverse drug interactions from ICSRs more precisely [101]. For that, it should be useful to take into account information available in the free text of the original reports [101]. Nevertheless, the lack of completeness of each report is the root of the problem because not all fields are required to be completed for reports to be accepted in VigiBase, and a detailed case-by-case analysis of each ICSR is needed [102].

## 5. Conclusions

Contrary to what was mentioned at the time of marketing, rivaroxaban has significant DDI potential with other drugs. Data analysis of VigiBase and some articles in this review highlight that PD interactions, as well as drugs that may impair haemostasis such as ASA or antithrombotics, are widely known and reported. Indeed, they occur due to the known properties of the drug and are predictable. However, this literature review shows that rivaroxaban has particular DDI potential with CYP3A/P-gp inhibitors and CYP3A/P-gp inducers, but the analysis of VigiBase data shows that the detection and reporting of pharmacokinetic interactions are sparse because they are not well recognised. Moreover, SmPC does not contain all potentially described post-marketing DDIs. This should serve as a warning to healthcare professionals as to the likelihood of occurrence of ADRs due to DDIs, as they are avoidable.

## Figures and Tables

**Figure 1 jpm-11-00250-f001:**
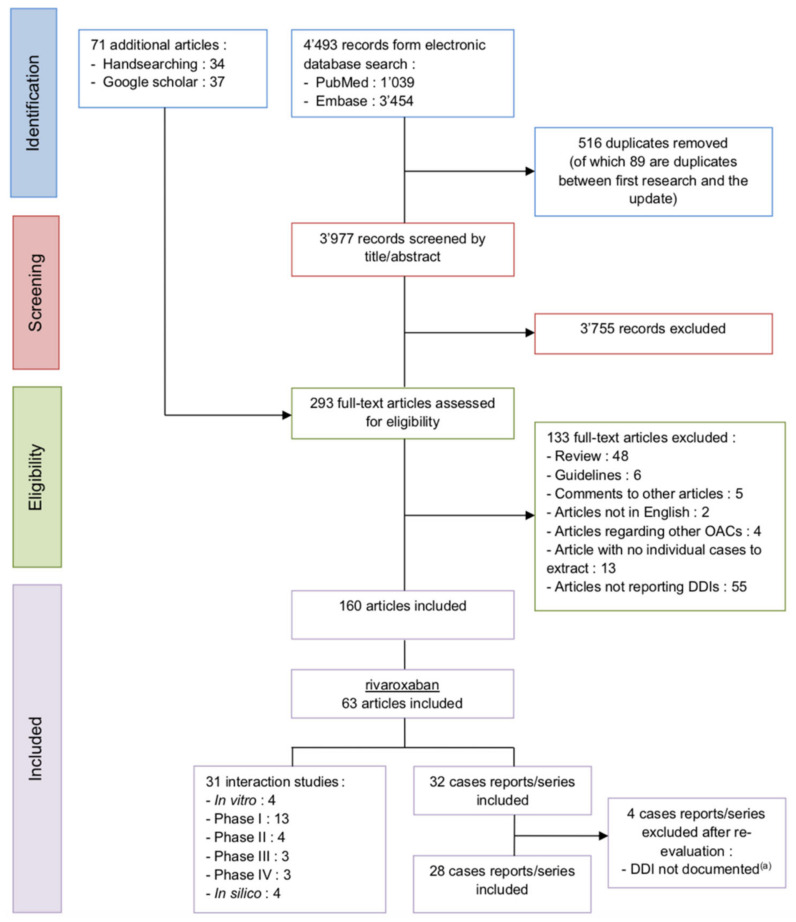
PRISMA flowchart of the rivaroxaban studies selection process DDI (drug-drug interaction) and OAC (oral anticoagulant), ^(a)^ or not understood from a pharmacological point of view.

**Table 1 jpm-11-00250-t001:** Eligibility criteria [25].

**Study Characteristics**	**Report Characteristics**
Type of studies In vitro and animal studies Randomised controlled trials Non-randomised studies Observational studies (including case series and case reports)	Language of publication English
Type of participants (human studies) Healthy subjects Patients under DOAC therapy for any pathology	Type of publications Published full-text articles and congress abstracts
Type of outcome Effect of potential interacting drugs on PK/PD profile of DOACs Effect of potential interacting drugs on DOACs safety profile: increase in the risk of haemorrhage or thromboembolic events Effects of DOACs on the PK/PD profile of potential interacting drugs	Year of publication From database inception to 20 August 2018 (PubMed, Embase) and from 2011 to 20 August 2018 (Google Scholar)

DOAC: direct oral anticoagulant/PD: pharmacodynamic/PK: pharmacokinetic.

**Table 2 jpm-11-00250-t002:** Summary of DDIs involving rivaroxaban.

Interactions Tested	Drugs Tested	References	Type of Study	Effect Observed
CYP3A/P-gp inhibitors	Amiodarone	[40]	Phase IV	↑ risk of bleeding
		[41]	Phase IV	↑ risk of major bleeding
		[42]	In silico	37% ↑ AUC
		[45]	In silico	×1.36 AUC
	Dronedarone	[41]	Phase IV	No increased risk of major bleeding
		[42]	In silico	31% ↑ AUC
	Clarithromycin	[33]	Phase I	54% ↑ AUC
		[34]	Phase I	94% ↑ AUC
		[41]	Phase IV	No increased risk of major bleeding
		[43]	In silico	×1.3 AUC
	Cyclosporine A	[37]	Phase II	102.6% ↑plasma levels
		[41]	Phase IV	No increased risk of major bleeding
	Erythromycin	[33]	Phase I	34% ↑ AUC
		[35]	Phase I	39% ↑ AUC
		[41]	Phase IV	N
	Diltiazem	[38]	Phase II	No significant increased risk of bleeding or thromboembolic event
		[41]	Phase IV	No increased risk of major bleeding
	Fluconazole	[33]	Phase I	42% ↑ AUC
		[41]	Phase IV	↑ risk of major bleeding
	Itraconazole	[41]	Phase IV	No increased risk of major bleeding
	Ketoconazole	[33]	Phase I	158% ↑ AUC
		[41]	Phase IV	No increased risk of major bleeding
		[43]	In silico	×2.3 AUC
	Non-DHP CCB	[39]	Phase III	No significant increased risk of thromboembolic event or clinically relevant bleeding ↑ risk of major bleeding or intracranial haemorrhage
	PDE5is	[32]	In vitro	↑ risk of bleeding
	Ritonavir	[33]	Phase I	153% ↑ AUC
		[43]	In silico	×2.2 AUC
	Tacrolimus	[62]	In vitro	No interaction
		[37]	Phase II	Plasma levels within therapeutic range (internal reference, 7–65 ng/mL)
	Verapamil	[36]	Phase I	38–41% ↑ AUC
		[41]	Phase IV	No increased risk of major bleeding
		[44]	In silico	48% ↑ AUC
	Voriconazole	[41]	Phase IV	No increased risk of major bleeding
CYP3A/P-gp inducers	Phenytoin	[41]	Phase IV	↑ risk of major bleeding
	Rifampicin	[41]	Phase IV	No increased risk of major bleeding
CYP3A/P-gp substrates	Atorvastatin	[41]	Phase IV	↓ risk of major bleeding
		[46]	Phase I	NCR effect
	Digoxin	[41]	Phase IV	No increased risk of major bleeding
		[46]	Phase I	NCR effect
	Midazolam	[33]	Phase I	NCR effect
Antithrombotic agents and NSAIDs	Aspirin	[47]	In vitro	↑ antithrombotic activity
		[52]	Phase I	↑ bleeding time
		[54]	Phase II	↑ risk of bleeding
		[55]	Phase II	No significant difference in the bleeding incidence
		[56]	Phase III	No increase in the risk of bleeding
		[57]	Phase III	↑ risk of all-cause death
		[58]	Phase IV	↑ risk of bleeding and ↑ risk of symptomatic thromboembolism
	Aspirin + clopidogrel	[47]	In vitro	↑ antithrombotic activity
	Aspirin + ticagrelor	[47]	In vitro	↑ antithrombotic activity
	Aspirin + thienopyridine	[54]	Phase II	↑ risk of bleeding
	Clopidogrel	[53] [55]	Phase I Phase II	↑ Bleeding time Significant decrease in the bleeding rate as compared to ticagrelor
	Enoxaparin	[48]	Phase I	50% ↑ anti-factor Xa activity
	Naproxen	[51]	Phase I	↑ bleeding time and 10% ↑ AUC
	NSAIDs	[56]	Phase III	No increased risk of bleeding (but limit of significance)
		[58]	Phase IV	↑ risk of bleeding
	Platelet aggregation inhibitor	[56]	Phase III	No increased risk of bleeding
		[58]	Phase IV	↑ risk of bleeding and ↑ risk of symptomatic thromboembolism
	Ticagrelor	[47]	In vitro	↑ antithrombotic activity
	Warfarin	[49]	Phase I	↑ PT/INR
		[50]	Phase I	↑ PT/INR
Gastric pH modifiers	Omeprazole	[60]	Phase I	NCR effect
	Ranitidine	[59]	Phase I	NCR effect
Other drugs	Irinotecan	[61]	In vitro	Inhibition of irinotecan active metabolite glucuronidation
	AS, CS, HA, klonopin, penicillin, TC, TA	[62]	In vitro	No effect

AS: alendronate sodium, AUC: area under the plasma concentration-time curve, CS: chondroitin sulphate, HA: hydrocodone-acetaminophen, INR: international normalised ratio, NCR: non-clinically relevant, PT: prothrombin time, TA: tranexamic acid, TC: tramadol chlorhydrate.

**Table 3 jpm-11-00250-t003:** Drug reported as interacting with rivaroxaban in VigiBase with interaction mechanism and most frequently reported adverse effect.

Drug B	No. of Occurrence	Mechanism	Mechanism Sub-Classification	Most Frequently Reported ADRs (No. Observed in Parenthesis)
Acetylsalicylic acid	48	PD	Additive pharmacological effect	Gastrointestinal haemorrhage (4838)
Alendronic acid	1	PD	Additive pharmacological effect	Upper gastrointestinal haemorrhage (4)
Alteplase	2	PD	Additive pharmacological effect	Haemorrhagic stroke (4)
Amiodarone	8	PK	Drug metabolism (inhibition)	Haemorrhage (46)
Apixaban	5	PD	Additive pharmacological effect	Gastrointestinal haemorrhage (102)
Azithromycin	2	PK	Drug metabolism (inhibition)	Pericardial haemorrhage (6)
Bosentan	1	PK	Drug metabolism (inhibition)	Anemia (3)
Carbamazepine	2	PK	Drug metabolism (induction)	Pulmonary embolism (6)
Celecoxib	8	PD	Additive pharmacological effect	Gastrointestinal haemorrhage (56)
Ciprofloxacin	1	PK	Drug metabolism (inhibition)	Blood urine present (3)
Citalopram	1	PD	Additive pharmacological effect	Melaena (7)
Clarithromycin	1	PK	Drug metabolism (inhibition)	Haemorrhage subcutaneous (4)
Clopidogrel	25	PD	Additive pharmacological effect	Gastrointestinal haaemorrhage (1009)
Dabigatran	1	PD	Additive pharmacological effect	Internal haemorrhage (18)
Dalteparin	2	PD	Additive pharmacological effect	Haemorrhagic anemia (3) Muscle haemorrhage (3)
Diclofenac	8	PD	Additive pharmacological effect	Gastrointestinal haemorrhage (40)
Dienogest/Ethinylestradiol	2	PD	Additive pharmacological effect	Menorrhagia (4)
Diltiazem	4	PK	Drug metabolism (inhibition)	Anemia (7)
Dipyrimadole	2	PD	Additive pharmacological effect	Cerebral haaemorrhage (3) Injection site haemorrhage (3)
Donepezil	2	PK	Drug metabolism (induction)	Cerebrovascular accident (3) Subdural haematoma (3)
Dronedarone	1	PK	Drug metabolism (inhibition)	Hematuria (6)
Drospirenone/ethinylestradiol	3	PD	Additive pharmacological effect	Deep vein thrombosis (6) Pulmonary embolism (6)
Duloxetine	1	PD	Additive pharmacological effect	Anemia (3)
Eicosapetaenoic acid	1	PD	Additive pharmacological effect	Haemorrhage subcutaneous (3)
Enoxaparin	15	PD	Additive pharmacological effect	Rectal haemorrhage (57)
Escitalopram	4	PD	Additive pharmacological effect	Haematoma (5)
Etodolac	2	PD	Additive pharmacological effect	Gastrointestinal haemorrhage (9)
Fluoxetine	2	PD	Additive pharmacological effect	Haematoma (4)
Fondaparinux	1	PD	Additive pharmacological effect	Haemorrhagic anemia (3)
Ginkgo biloba	3	PD	Additive pharmacological effect	Upper gastrointestinal haemorrhage (4)
Heparin	12	PD	Additive pharmacological effect	Rectal haaemorrhage (22)
Ibrutinib	3	PK/PD	Drug metabolism (inhibition) + additive pharmacological effect	Contusion (16)
Ibuprofen	16	PD	Additive pharmacological effect	Gastrointestinal haemorrhage (161)
Iloprost	1	PD	Additive pharmacological effect	Haemorrhage (4)
Indometacin	5	PD	Additive pharmacological effect	Gastrointestinal haemorrhage (12)
Itraconazole	2	PK	Drug metabolism (inhibition)	Ecchymosis (4) Epistaxis (4)
Ketoprofen	1	PD	Additive pharmacological effect	Anemia (9)
Ketorolac	2	PD	Additive pharmacological effect	Contusion (4)
Lenalidomide	1	PD	Additive pharmacological effect	Epistaxis (5)
Levonorgestrel	3	PD	Additive pharmacological effect	Menorrhagia (11)
Losartan	1	PK	Drug metabolism (inhibition)	Haemoglobin decreased (9)
Loxoprofen	1	PD	Additive pharmacological effect	Gastric ulcer haemorrhage (4)
Lubiprostone	1	PD	Additive pharmacological effect	Gastrointestinal haemorrhage (3)
Meloxicam	6	PD	Additive pharmacological effect	Gastrointestinal haemorrhage (70)
Metamizole	1	PD	Additive pharmacological effect	Upper gastrointestinal haemorrhage
Methylprednisolone	1	PD	Additive pharmacological effect	Anemia (3)
Nabumetone	1	PD	Additive pharmacological effect	Upper gastrointestinal haemorrhage (3)
Nadroparin	1	PD	Additive pharmacological effect	Hematuria (4)
Naproxen	11	PD	Additive pharmacological effect	Gastrointestinal haemorrhage (135)
Paroxetine	3	PD	Additive pharmacological effect	Anemia (5)
Phenprocoumon	3	PD	Additive pharmacological effect	Hematochezia (4) Intestinal haemorrhage (4)
Pomalidomide	1	PK	Drug metabolism (inhibition)	Gastrointestinal haemorrhage (3)
Prasugrel	7	PD	Additive pharmacological effect	Gastrointestinal haemorrhage (37)
Prednisolone	5	PD	Additive pharmacological effect	Anemia (5)
Prednisone	6	PD	Additive pharmacological effect	Gastrointestinal haemorrhage (19)
Rifampicin	1	PK	Drug metabolism (induction)	Pulmonary embolism (8)
Riociguat	8	PD	Additive pharmacological effect	Epistaxis (30)
Sertraline	2	PD	Additive pharmacological effect	Anemia (4)
Sorafenib	1	PD	Additive pharmacological effect	Epistaxis (4)
Streptokinase	1	PD	Additive pharmacological effect	Haemorrhage (3)
Sunitinib	2	PD	Additive pharmacological effect	Gastrointestinal haemorrhage (6)
Tadalafil	1	PK	Drug metabolism (inhibition)	Haemorrhage (4)
Ticagrelor	5	PD	Additive pharmacological effect	Gastrointestinal haemorrhage (26)
Treprostinil	6	PD	Additive pharmacological effect	Haemorrhage (13)
Venlafaxine	2	PD	Additive pharmacological effect	Epistaxis (5)
Verapamil	2	PK	Drug metabolism (inhibition)	Haemoglobin decreased (3) Anemia (3)
Warfarin	21	PD	Additive pharmacological effect	Gastrointestinal haemorrhage (406)

## Data Availability

The data presented in this study are available on request from the corresponding author.

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
