# Peer review of "Drug-Drug Interactions Leading to Adverse Drug Reactions with Rivaroxaban: A Systematic Review of the Literature and Analysis of VigiBase"

_jpm, 2021, doi:10.3390/jpm11040250_

Round 1

Reviewer 1 Report

Introduction

Line 36-38: the reference num 4 says "to take higher doses of rivaroxaban (15 or 20 mg tablets) with food to ensure optimal drug absorption" and apixaban doses <60kg are different, so not in the line with the sentence: less impacted by food or bodyweight.

Line 42, reference 5 after dabigatran is for the SmPC of Edoxaban.

Line 50: Please consider revising - statement suggests no monitoring is required but then suggests monitoring.  Please add reference.

Methods

Line 135, the reference for WHODrug is missing.

Line 158. They use the highest MedDRA hierarchy SOC which can be really broad. Another approach could have been a MedDRA SMQ (Standardized MedDRA Query, which are groupings of MedDRA PTs related to a defined medical condition or area of interest). There’s one MedDRA SMQ GI bleeding: Gastrointestinal perforation, ulceration, haemorrhage or obstruction, that could have been used. GI disorders could be Dental and gingival conditions, Diverticular disorders, Malignant Neoplasm…among others.  Please consider the above or provide a rationale not using more detailed term types.

Results

Table 3.

It is unclear what the numbers in the parenthesis represent. Please provide greater details.  What is number of occurrence?

Discussion

Line 450 reference missing.

Line 477: venous thromboembolism is the most frequently reported ADR? This is more of an indication for the drug. Please clarify if this is an ADR or reason for use?

Line 500: the authors state the knowledge of health professionals is the same as from patients… poor patients.

References

I’d suggest referencing pharmacology books when explaining drug mechanism of action. Suggested reference for 1 and 2 references: Weitz JI. Blood coagulation and anticoagulant, fibrinolytic, and antiplatelet drugs. In: Goodman & Gilman’s The pharmacological basis of therapeutics. Brunton LL (Ed). 12th edition. McGraw Hill.

Some paragraphs have 2 or 3 sentences from the same reference and this is state in each end of sentence, is that correct? (lines 191-196)

Author Response

Introduction

Line 36-38: the reference num 4 says "to take higher doses of rivaroxaban (15 or 20 mg tablets) with food to ensure optimal drug absorption" and apixaban doses <60kg are different, so not in the line with the sentence: less impacted by food or bodyweight.

We thank the reviewer for the comment and agree that absorption of high dose of rivaroxaban is dependent on food intake and that apixaban doses should be lowered in patients < 60 kg. However, we meant in the sentence “less impacted by food or body weight”, that NOACs are less influenced by food or body weight than AVK, which is a benefit in terms of ease of use. Indeed, this reference points out that “Unlike warfarin, which has a narrow therapeutic window and requires individualized dosing based on the international normalized ratio (INR), the NOACs have a wide therapeutic window, thereby facilitating fixed dosing in adults without the need for laboratory monitoring or dose adjustments for body weight” and “warfarin is associated with > 10-fold interindividual variations in dose to achieve therapeutic anticoagulation. Its pharmacokinetics and pharmacodynamics are influenced by genetic polymorphisms (CYP 2C9 and VKORC1), dietary vitamin K intake, concomitant medications, alcohol use, patient age, body weight, and various disease states, necessitating regular coagulation monitoring to ensure that the patient's INR remains within the target range.”

The sentence was reformulated (line 38 and line 40-43)

“Indeed, DOACs are considered easier to use because they have a wide therapeutic window, less interindividual variability, and higher oral bioavailability that is less impacted by food intake or body weight than warfarin, the reference treatment [3]. They thus no longer need to be individualized on a daily basis like VKAs, which require monitoring of the international normalized ratio (INR) [3]. However, although DOACs are less influenced by food or body weight, small dose adjustments are necessary for high-dose rivaroxaban and for low-weight patients < 60 kg taking apixaban [3].”

Line 42, reference 5 after dabigatran is for the SmPC of Edoxaban.

We apologize for this error and have changed the reference accordingly.

Line 50: Please consider revising - statement suggests no monitoring is required but then suggests monitoring.  Please add reference.

We thank the reviewer for the comment, and we have revised to sentence (line 55-57). Indeed, no monitoring is required to prescribe rivaroxaban, and there are no specific laboratory parameters available to monitor factor Xa inhibitors but factor antiXa could be used if the measurement is necessary, such as to confirm an overdose,

“There is therefore no additional need for a priori monitoring of specific laboratory parameters, but anti-Xa factor could be used in specific cases where a measurement is needed, for example to confirm an overdose [13].”

Methods

Line 135, the reference for WHODrug is missing.

The reference for WHODrug was added (line 149).

Line 158. They use the highest MedDRA hierarchy SOC which can be really broad. Another approach could have been a MedDRA SMQ (Standardized MedDRA Query, which are groupings of MedDRA PTs related to a defined medical condition or area of interest). There’s one MedDRA SMQ GI bleeding: Gastrointestinal perforation, ulceration, haemorrhage or obstruction, that could have been used. GI disorders could be Dental and gingival conditions, Diverticular disorders, Malignant Neoplasm…among others.  Please consider the above or provide a rationale not using more detailed term types.

We thank the reviewer for the comment. We initially thought about using the SMQ, but after discussion with Uppsala Monitoring Center, it was decided that SOC would be the best approach. Indeed, we were not looking for simple ADRs collection, but we had to add a third parameter as we were looking for ADRs due to drug-drug interactions. We anticipated that we would extract a lot of bleeding ADRs, but we were also interested in seeing if anything else that was not in the SMQ you mentioned would come out of this search. Broader SOC search therefore seemed more suitable.

Results

Table 3.

It is unclear what the numbers in the parenthesis represent. Please provide greater details.  What is number of occurrence?

The numbers in the parentheses represent the number of times the most reported adverse drug reaction for a pair of drugs were reported. An explanation was added in the manuscript to avoid misunderstanding (line 493-496).

The “No. of occurrences” is the number of different verified adverse drug reactions reported for a pair of drugs. For example, the interaction between rivaroxaban and AAS led to 48 different categories of adverse drug reactions.

“Concerning verified PK DDIs, inhibitors of CYP3A and P-gp were the most reported drugs, and bleeding was the most reported ADR (Table 3). Regarding verified PD DDIs, antithrombotic agents and NSAIDs were the most reported drugs, and bleeding was also the most reported ADR. Regarding bleeding, the most reported site was the gastrointestinal tract (Table 3). Table 3 shows the number of occurrences that represents the number of different ADRs that occurred after the interaction between rivaroxaban and drug B, and the number in parentheses is the number of the most frequently reported ADR.”

Discussion

Line 450 reference missing.

References were added (line 506).

Line 477: venous thromboembolism is the most frequently reported ADR? This is more of an indication for the drug. Please clarify if this is an ADR or reason for use?

The discussion here refers to the ADRs reported in the case reports included in our literature search. Venous thromboembolism was the second most reported ADR in these case reports, and not mentioned in rivaroxaban’s SmPC. We compared ADRs reported in published case reports with the SmPC to verify the coverage of our literature search. Venous thromboembolism might not be recognized as an ADR in the SmPC but rather as a lack of efficacy. Sentence was changed to clarify. Moreover, it is not the most reported ADR but one of the most reported ADR, and we have changed the sentence accordingly (line 531-535).

Venous thromboembolism was identified in the case reports included in our literature search as one of the most frequently reported ADR of rivaroxaban, and it was not mentioned, per se, in rivaroxaban’s SmPC. This is likely due to the fact that interactions leading to this ADR are not recognized and are instead classified as treatment inefficacy  [20]. Therefore, this is not a lack of coverage in our literature search.”

Line 500: the authors state the knowledge of health professionals is the same as from patients… poor patients.

We changed the sentence accordingly. VigiBase is a database based on spontaneous reporting and drug-drug interaction and subsequent ADRs were reported by both health professionals and patients. There is overall an underreporting of PK DDIs as compared to PD DDIs by both health professionals and patients which might be caused by the lack of understand of PK DDIs (line 565-569).

“Indeed, this emphasizes a bias in the data included in VigiBase, which depends on spontaneous reporting. As healthcare professionals and/or patients are the source of these spontaneous reports and as they often less familiar with PK DDIs, these are underreported because they go undetected. These results are consistent with those of a study that used the same database, where PD and PK DDIs accounted for 41% and 25% of DDIs, respectively [98].”

References

I’d suggest referencing pharmacology books when explaining drug mechanism of action. Suggested reference for 1 and 2 references: Weitz JI. Blood coagulation and anticoagulant, fibrinolytic, and antiplatelet drugs. In: Goodman & Gilman’s The pharmacological basis of therapeutics. Brunton LL (Ed). 12th edition. McGraw Hill.

The references 1 and 2 were replaced at Line 32 by the pharmacology book that you suggested. The previous reference 1 was left at Line 57 because it did not explain a drug mechanism of action.

Some paragraphs have 2 or 3 sentences from the same reference and this is state in each end of sentence, is that correct? (lines 191-196)

Yes, this is correct. This reference includes multiple CYP3A inhibitors.

Reviewer 2 Report

Authors performed systematic review of published articles and case reports, and summarized outcomes with respect to  drug-drug interactions of oral anticoagulant rivaroxaban. The procedures and analysis employed were explained adequately. The manuscript provides a nice summary of pharmacokinetic and pharmacodynamic DDIs and adverse reactions from about 60 plus publications, and which are clinically relevant. Overall manuscript is well written.

Author Response

Authors performed systematic review of published articles and case reports, and summarized outcomes with respect to drug-drug interactions of oral anticoagulant rivaroxaban. The procedures and analysis employed were explained adequately. The manuscript provides a nice summary of pharmacokinetic and pharmacodynamic DDIs and adverse reactions from about 60 plus publications, and which are clinically relevant. Overall manuscript is well written.

We thank the reviewer for these favorable comments.

Reviewer 3 Report

Authors provided the careful and detailed review of ADR induced by rivaroxaban interactions with other drugs.  

Not surprisingly, they could show that rivaroxaban and ASA/ antithrombotics have additive pharmacological effect resulting in GI hemorrhage. The other conclusion is that rivaroxaban has particular DDI potential with CYP3A/P-gp inhibitors and CYP3A/P-522 gp inducers, but not recognized in VigiBase. Databases depending on spontaneous reporting (such as VigiBase) have many disadvantages, but they are clearly recognized in the paper.

Comments:

- The presented data are comprehensive and valuable, however their scientific soundness is moderate.

- To what extend this paper fits to aims and scope of Journal of Personalized Medicine? (up to the editor's decision)

- Obvious shortcoming is that data from VigiBase were retrieved in early 2018, thus 3 years ago.

Minor remarks

- Small differences in searching strategies in 3 databases (page 2, from line 90.) – any reason for that or just unintentional?

- In table 2, some opposite effects regarding the same DDI are presented (e.g. No increased risk of bleeding VS increased risk of bleeding - for NSAIDs). Short comment on interpretation of discrepancies between particular studies would be useful for readers.

- Exact meaning of “No. of occurrence” in Tab 3?

Author Response

Authors provided the careful and detailed review of ADR induced by rivaroxaban interactions with other drugs.  

Not surprisingly, they could show that rivaroxaban and ASA/ antithrombotics have additive pharmacological effect resulting in GI hemorrhage. The other conclusion is that rivaroxaban has particular DDI potential with CYP3A/P-gp inhibitors and CYP3A/P-gp inducers, but not recognized in VigiBase. Databases depending on spontaneous reporting (such as VigiBase) have many disadvantages, but they are clearly recognized in the paper.

Comments:

- The presented data are comprehensive and valuable, however their scientific soundness is moderate.

We thank the reviewer for this comment on our work, and we agree that much work remains to be done to make VigiBase data more robust. A future meta-analysis could assess the published data as a whole and the clinical significance.

- To what extend this paper fits to aims and scope of Journal of Personalized Medicine? (up to the editor's decision)

As explained in the “Special Issue Information”, the use of personalized medicine is the future of clinical practice, but its implementation is hindered by many factors, such as the lack of healthcare professionals’ teaching and information. Our review summarizes the pharmacokinetic and pharmacodynamic drug-drug interactions involving rivaroxaban, and allows to better understand, and thus better predict, the complexity and interplay between sources of variability in efficacy and safety of rivaroxaban treatment.

- Obvious shortcoming is that data from VigiBase were retrieved in early 2018, thus 3 years ago.

We agree with the reviewer that this a shortcoming of our review. It appears complicated to get updated data as the database has to be frozen and cleaned by the Uppsala Monitoring Center. Moreover, we believe that our current analysis already contains a lot of information regarding adverse drug reactions involving rivaroxaban. It highlights the lack of awareness on pharmacokinetics drug-drug interactions by spontaneous reporters and that most reported adverse drug reactions were bleeding events. The inherent, and mentioned, limitations of this type of database would remain regardless of whether we would have added 3 additional years.

Minor remarks

- Small differences in searching strategies in 3 databases (page 2, from line 90.) – any reason for that or just unintentional?

They are indeed small differences in the search strategy in the three databases. For Embase, we used exactly the same strategy built for PubMed in the first instance. However, the combination of keywords yielded a much higher number of results and most of them were not relevant to our search. It was thus necessary to adapt the search strategy to Embase and the term “interaction” was thus omitted, obtaining a substantial reduction in the number of results retrieved. The search strategy was also adapted for Google Scholar, as it is a search engine and not a database. Moreover, it does not allow the use of MeSH (Medical Subject Heading) terms and the use of Boolean operators is more limited than in PubMed or Embase. Additionally, in Google Scholar, the singular form of a term is not assumed when typing its plural form, so in our query it was essential to include both terms “interaction” and “interactions”. We also decided to include only the common name of each DOAC to simplify the search strategy and to incorporate the keyword “case report” because Google Scholar was employed for the specific purpose of gathering additional relevant case reports or case series.

- In table 2, some opposite effects regarding the same DDI are presented (e.g. No increased risk of bleeding VS increased risk of bleeding - for NSAIDs). Short comment on interpretation of discrepancies between particular studies would be useful for readers.

We thank the reviewer for the comment. We have rechecked all the discrepant results and modified for “No increased risk of major bleeding” when the results did not reach statistical significance. Some discrepancies however remained and possible explanations have been provided in the results’ section.

“In contrast, the coadministration of rivaroxaban and erythromycin or clarithromycin decreased the risk of major bleeding but it was not statistically significant [41]. Coadministration of rivaroxaban with cyclosporin, verapamil, diltiazem, ketoconazole and itraconazole, voriconazole, posaconazole and dronedarone did not significantly change the incidence rate of major bleeding [41]. Results for erythromycin, clarithromycin, cyclosporin, verapamil, ketoconazole and dronedarone are not in line with previous cited studies. This could be because this study has strong limitations of a retrospective design and of an analysis based on Health Insurance database system, and thus a lack of detailed clinical information, such as liver and kidney fonction [41]. Moreover, this study included an Asian population that has been recognized to have a different bleeding risk and anticoagulant therapy than the Western population [41]. Finally, rivaroxaban dosage and concomitant treatment were not considered in the model [41].”

“Coadministration of rivaroxaban with rifampicin and phenytoin was assessed and surprisingly showed an increased risk of major bleeding [41]. However, this effect was not statistically significant for rifampicin [41]. Phenytoin, as a CYP inducer, is expected to decrease rivaroxaban AUC and, therefore, the bleeding risk. The results of this phase IV study should be treated with caution due to the limitations mentioned above [41].”

“The bleeding risk with rivaroxaban was assessed when coadministered with atorvastatin and digoxin and significantly decreased risk of major bleeding was observed, while the effect of digoxin was not statistically significant [41]. In the phase I study, atorvastatin had no effect on rivaroxaban PK and this discrepancy in results can also be attributed to the limitations of the phase IV study [41].”

“In a sub-analysis of pooled data from the RECORD programme, coadministration of NSAIDs (relative rate ratio = 1.22, CI95% = 0.99-1.50), platelet aggregation inhibitors (PAIs) and ASA (relative ratio = 1.32, CI95% = 0.85-2.05) together with rivaroxaban increased the risk of bleeding in hip or knee replacement surgery patients, although the effect was not considered significant [56]. However, the small proportion of patients using concomitant PAIs and ASA may not have been high enough to conclude on the risk of bleeding, which could explain the difference in results with other studies [56]. Regarding the increased risk of bleeding with concomitant use of NSAIDs, it was at the limit of statistical significance [56].”

“In a sub-analysis of the XAMOS study, coadministration of PAIs (including ASA) increased the incidence of symptomatic thromboembolic and bleeding events in patients taking rivaroxaban and in those who followed standard thrombophylaxis for VTE prophylaxis after major orthopaedic surgery [58]. However, this finding was largely attributable to a higher incidence of symptomatic arterial thromboembolic events [58]. This could be explained by the fact that PAIs users were older and had more comorbidities affecting cardiovascular risk [58].” 

- Exact meaning of “No. of occurrence” in Tab 3?

The “No. of occurrences” is the number of different verified adverse drug reactions reported for a pair of drugs. For example, the interaction between rivaroxaban and AAS led to 48 different categories of adverse drug reactions
